



Atmospheric Chemistry and Physics
2020 Manuscript, Rev. May 2020

**Captured Cirrus Ice Particles in High Definition**

Nathan Magee*, Katie Boaggio, Samantha Staskiewicz, Aaron Lynn, Xuanyi Zhao, Nicholas
Tusay, Terance Schuh, Manisha Bandamede, Lucas Bancroft, David Connelly, Kevin Hurler,
Bryan Miner, and Elissa Khoudary.

*Corresponding Author:  magee@tcnj.edu

**Affiliations:**
Boaggio:  ORISE Participant at U.S. Environmental Protection Agency
Hurler:  University of South Carolina
Bandamede: Ross University School of Medicine
Connelly: Cornell University
Bancroft: Universal Display Corporation
Staskiewicz: The Pennsylvania State University
Magee and others: The College of New Jersey (TCNJ)



**Abstract**
Cirrus clouds composed of small ice crystals are often the first solid matter encountered by
sunlight as it streams into Earth's atmosphere. A broad array of recent research has emphasized
that photon-particle scattering calculations are very sensitive to ice particle morphology,
complexity, and surface roughness. Uncertain variations in these parameters have major
implications for successfully parameterizing the radiative ramifications of cirrus clouds in
climate models. To date, characterization of the microscale details of cirrus particle morphology
has been limited by the particles' inaccessibility and technical difficulty in capturing imagery
with sufficient resolution. Results from a new experimental system achieve much higher
resolution images of cirrus ice particles than existing airborne particle imaging systems. The
novel system (Ice Cryo-Encapsulation by Balloon, ICE-Ball) employs a balloon-borne payload
with environmental sensors and hermetically-sealed cryo-encapsulation cells. The payload
captures ice particles from cirrus clouds, seals them, and returns them via parachute for vapor-
locked transfer onto a cryo-scanning electron microscopy stage (cryo-SEM). From 2016-2019,
the ICE-Ball system has successfully yielded high resolution particle images on nine cirrus-
penetrating flights. On several flights, including one highlighted here in detail, thousands of
cirrus particles were retrieved and imaged, revealing unanticipated particle morphologies,
extensive habit heterogeneity, multiple scales of mesoscale roughening, a wide array of
embedded aerosol particles, and even greater complexity than expected.



## 1. Introduction

Understanding of cirrus cloud microphysics has advanced dramatically in the past several
decades thanks to continual technical innovations in satellite remote sensing, in-situ aircraft
measurements, sophisticated laboratory experiments, and modeling that incorporates this new
wealth of data. In combination, the au courant picture of cirrus clouds has emerged: a highly
complex system that results in a vast array of cirrus formations, varying in time and location
through interdependent mechanisms of microphysics, chemistry, dynamics, and radiation (e.g.
Heymsfield et al. 2017). While the net magnitude of cirrus radiative forcing is clearly not as
large as thick low-altitude clouds, an intricate picture of climate impacts from cirrus is coming
into focus. It now seems clear that both the sign (positive or negative) and strength of cirrus
radiative forcings and feedbacks depend on variables that can change with a wide array of
parameters: geography, season, time of day, dynamical setting, and the concentrations, shapes,
sizes, and textures of the cirrus ice particles themselves (e.g. Burkhardt and Kärcher, 2011;
Harrington et al. 2009;  Järvinen 2018b, Yi et al. 2016). Furthermore, many of these factors may
be changing markedly over time, as contrail-induced cirrus and changing temperature, humidity,
aerosol in the high troposphere are affected by evolving anthropogenic influences (Randel and
Jensen, 2013; Kärcher et al. 2018, Zhang et al. 2019). Undoubtedly, a sophisticated, high-
resolution understanding of cirrus is critical to accurately model the impacts to global and
regional climate.

40          Satellite-derived measurements of cirrus properties have become vastly more

sophisticated with the advent of increased spatial and temporal resolution, a broader array of
spectral channels, specialized detectors, and advances in scattering theory (e.g. Yang 2008;
Baum 2011; Sun 2011; Mauno 2011; Yang 2013; Cole 2014; Tang 2017, Yang et al. 2018).
Where a generation ago it was challenging to even isolate the presence of cirrus clouds in much
satellite imagery, it is now routine to derive estimates of ice cloud optical depth, cloud top
temperature, cloud top height, effective particle size, and in some cases even to infer the
dominant particle habit and roughness of crystal surfaces (McFarlane 2008; King 2013; Cole
2014, Hioki et al. 2016, Saito et al. 2017). The emerging ubiquity of this sophisticated satellite
data and highly-developed retrieval schemes can sometimes obscure the fact that major
fundamental uncertainties remain regarding cirrus microphysical compositions and their
intertwined dynamic evolution.




53        Cloud particle imaging probes on research aircraft have also contributed to major leaps in

understanding, helping to constrain cirrus property satellite retrievals and climate modeling
representations  (Baumgarnder et al. 2017; Lawson et al. 2019). These probes deliver particle
imaging and concentration measurements that yield unique insights into ice particle habits and
distributions in cirrus, though several significant limitations remain. The SPEC Inc. CPI probes
have flown for nearly 20 years and can achieve 5 μm particle optical resolutions and SPEC's 2D-
S stereo imaging probe yields 10 μm pixel sizes (Lawson et al. 2019). For example, CPI images
of cirrus ice were featured on the June 2001 cover of the Bulletin of the American
Meteorological Society (Connelly et al. 2007) and have contributed to many other cloud physics
field programs since (for complete list, see Appendix A in Lawson et al. 2019). Other recent in-
situ particle measurement innovations include the HOLODEC (Fugal 2004), SID3 (Ulanowski et
al. 2012, Järvinen et al 2018a), and PHIPS-Halo (Schnaiter 2018), with imaging resolutions on
the order of 5-10 microns, as well as multi-angle projections, and indirect scattering
measurements of particle roughness and complexity. High speed aerodynamics and concerns
about instrument-induced crystal shattering have produced some uncertainties regarding inferred
particle concentrations, size distributions, and orientations, but perhaps more importantly, the
limited optical resolving power means that in-situ imaging instruments are not able to determine
fine-scale details of crystal facet roughness or highly complex habit geometry, particularly for
small ice crystals. Several groups have also achieved recent in-situ measurements of cirrus
particles using balloon-borne instruments (Miloshevich and Heymsfield 1997; Cirisan et al.
2014; Kuhn and Heymsfield 2016; Wolf et al. 2018). Though this has been a relatively sparse
set, some slight momentum appears to be building toward exploiting advantages of this slower-
speed probe.

76        The synthesis that has been emerging describes cirrus clouds that are often, but not

always, dominated by combination of complex particle morphologies, and with crystal facets that
usually show high roughening and complexity at the microscale (Baum et al. 2011; Yang et al.
2013; Yi et al. 2013; Tang et al. 2017; Heymsfield et al. 2017; Lawson et al. 2019). Particle
complexity has been considered to encompass an array of potential geometric deviations away
from a simple hexagonal, single ice crystal:  intricate polycrystalline morphological shapes,
aggregations of individual particles, partial sublimation of particles, post-sublimation regrowth



of microfacets, and inclusions of bubbles and aerosol particles (Ulanowski et al. 2012; Schnaiter
et al. 2016; Voitlander et al. 2018). Even where crystals may present mainly planar facet
surfaces, these surfaces are often characterized by regular or irregular patterns of roughening at
multiple scales. All aspects of increased complexity and roughening have been shown to smooth
and dampen the characteristic peaks in the scattering phase function of hexagonal ice crystals
(van Diedenhoven 2014). The angular integral of the phase function yields the asymmetry
parameter, which has been broadly applied as an indicator of net radiative impact of underlying
particle microphysics (Baran 2015). With mesoscopic crystal roughness and complexity
contributing to less total forward scattering, the asymmetry parameter and net downwelling
radiation is reduced (e.g. Yang and Liou 1998; Um and McFarquhar 2011, van Diedenhoven et
al. 2013). The calculated impacts on cirrus cloud radiative effect are shown to be
climatologically significant compared to assumptions that cirrus composed of less complex
crystals (Yang et al. 2013; Järvinen et al. 2018b). Furthermore, beyond questions of particle
morphology and radiative balances, major uncertainties around cirrus cloud evolution remain
regarding particle nucleation pathways and the interconnected roles of aerosol chemistry, high-
altitude humidity, and the subtle dynamics of vertical motion and turbulent eddies in cirrus.
**2.  ICE-Ball in-Situ Capture Methods**
**2.1 ICE-Ball System**
The ICE-Ball experiment has been designed, refined, and implemented from 2014-2018. The
basic system consists of a ~2 kg payload ("Crystal Catcher") carried aloft by a 300 g latex
weather balloon. The payload components are enclosed in a mylar-wrapped Styrofoam cube
(Fig. 1) to prevent electronics from freezing and to comply with FAA regulations for weight,
density, and visibility.  Figure 1 shows authors Tusay, Lynn, and Zhao holding ICE-Ball, along
with a cross-section diagram of the cryo-collection and preservation mechanism. The instrument
suite consists of standard balloon sonde sensors (pressure, temperature, and dewpoint), and also
includes HD video (GoPro Session) and dual real-time GPS position tracking (SPOT and
GreenAlp). The cryo-capture vessel and ice encapsulation cell comprise the novel ice particle
capture and preservation mechanism. Several versions of this mechanism have been employed,
but in each case, it has consisted of a vacuum-insulated stainless steel vessel (250 ml volume)
filled with crushed dry ice and containing a custom-machined sweep tube and ice encapsulation
cell. The sweep tube extends slightly above the top of the payload, and passively collects





particles in its path due to the upward motion of the balloon (~5 m/s). When the collection
aperature is open, the particles settle to the bottom of the collection tube and are gravitationally
deposited in the ice encapsulation cell, which is nestled in the surrounding dry ice. The
encapsulation cell interior diameter is 7 mm, and has an open volume of 0.2 cm$^3$.
During ascent, the balloon is ~20 ft above the payload and does not appear to affect
particle concentrations impacting the top of the payload. Several sweep tube geometries and
opening sizes have been tested (from 0.5 to 5 cm$^2$), but in each case, streamline modeling and
sample analyses suggest that collection efficiencies are high for particles larger than 50 microns
and decrease at smaller sizes.  Cirrus cloud conditions and the in-flight collection operation is
recorded via the go-Pro video.  Cirrus particles are routinely observed passing the camera, and
either 22° halos and/or circumzenith arcs can often be observed on the video record of each
successful flight.
**2.2 Ice Crystal Preservation**
The apertures to the cryo-vessels' sweep-tubes can be opened and closed using a rotational servo
motor that is driven by an Arduino microprocessor (a previous version used robotic clamshell
seals, as seen in Supplement2 video). The Arduino is programmed to open the path to each
collection vessel individually at cirrus altitudes that are prescribed before each launch.
Immediately after transiting the prescribed collection zone(s), the apertures are closed and a
magnetic sphere is dropped down the collection tube to seal the collected crystals in the small-
volume encapsulation cell (see Fig. 1b). This onboard preservation system has been tested to
preserve collected crystals in pristine condition for approximately 6 hours, which usually
provides ample time for recovery. Upon ICE-Ball landing and recovery, the small volume
encapsulation cell is hermetically double-sealed and stored in dry ice to ensure that crystals are
preserved as pristinely as possible.  After returning to the lab, the sealed ice-crystal samples can
also be stored in liquid nitrogen for medium-term storage of up to several days prior to transfer
and imaging in the cryo-SEM.
**2.3 Flight Record**
Intensive field campaigns were conducted during June and July of 2016-2019, consisting of 5-10
flights per campaign. In order to proceed with mission launch, the following conditions were
required:  1) greater than 50% projected cirrus coverage at the time of launch, 2) horizontal wind
speeds (trajectory mean) less than 60 kt, 3) modeled trajectory allowing for a safe launch zone


and an open landing zone within a 1 hour drive of TCNJ, 4) FAA/ATC approval, requiring flight
plan filing 24 hours prior to launch. Conditions that prevented launches on particular days
mainly included high wind speeds at altitude, and clear skies or poorly predicted cirrus cloud
coverage. During mid-Atlantic summer, high altitude mean wind speeds meet the speed 60 kt
maximum launch threshold approximately 60% of the time; regional climatological proximity to
the jetstream often results in prohibitively high winds in the upper troposphere during other
seasons. High wind speeds result in a longer flight trajectory (50 kt mean wind yields ~ 50 mile
flight), degrading landing zone accuracy (nominal landing position prediction error radius of
10% of the trajectory length). Longer flight paths also require additional drive time and increase
the risk of landing in an inaccessible or unsafe location (e.g. Atlantic Ocean, Military Base,
Airport, or Interstate). In the summertime mid-Atlantic region, cirrus coverage is approximately
20%. The accuracy of cirrus coverage forecasts by NCEP operation weather models (GFS,
NAM, and HRRR) were found to be a significant challenge to launch planning. Models of high-
cloud forecasts appear not to produce significant skill beyond ~48 hour lead times, thought it is
likely that these fields have not been refined as carefully as others due to modest influence on
surface weather.
The novel experimental system has failed to recover ice crystals on more occasions than
it succeeded (38% crystal recovery rate). As the team gained more experience, the success rate
improved (65% during the final campaign), but systemic experimental challenges remain.
Conditions that resulted in failure to capture or recover cirrus ice crystals were somewhat varied:
system technical failures including premature balloon bursts and frozen electronics (6
occurrences); ICE-BALL landing zone (often high in a tree canopy) resulted in recovery time
that was too long to preserve crystals (6 times); flight trajectory missed scattered cirrus clouds (4
times); failure of Cryo-transfer or SEM outage (2 times). Perhaps the most difficult obstacle to
the further development and deployment of the experimental system is the challenge associated
with difficult to access landing zones. This is especially challenging in the mid-Atlantic where
geography results in only small pockets of public property and high fractions of tree coverage.
Remarkably, all 28 flights were eventually recovered, but 4 of these included instances of the
system caught higher than 50 feet up in a tree, which typically resulted in a complex multi-day
effort to retrieve.


### 2.4. Vapor-lock transfer and cryo-SEM imaging

**2.4. Vapor-lock transfer and cryo-SEM imaging**
A unique cryo-SEM imaging capability for captured samples is provided by a Hitachi SU5000
SEM, equipped with a Quorum 3010 Cryosystem and EDAX Octane Enegry Dispersive
Spectroscopy (EDS).  The Hitachi SU5000 is employs a Schottky field emission electron gun
and variable pressure sample chamber.  The combination of the variable-pressure FE-SEM
chamber with the Quorum cryosystem is a unique configuration that allows samples to be
transferred, held, and imaged uncoated at very low temperature (usually ~160°C), while
simultaneously ensuring that excess water vapor is not deposited or removed from the sample
surfaces.  The Quorum 3010 Cryosystem integrates a cryo-airlock that transfers a frozen
encapusulation cell into the SEM chamber while maintaining cryo-cooling and hermetic sealing
throughout the transfer process.  Once the SEM chamber has been loaded with the crystal sample
and balanced cryo-temperature and pressure achieved, the magnetic seal is removed and imaging
can commence.

188       Electron beam accelerations of 2kV – 20 kV have been successfully employed with

Hitachi backscatter and secondary electron detectors to produce micrographs of the captured ice
crystals. The backscatter images in particular produce a dramatic contrast between the ice and
higher-density embedded aerosol particles that often include silica minerals and metal oxides.
The image resolutions of individual micrographs depend on multiple factors including SEM
beam energy, spot size, working distance, and beam scanning speed.  Generally, lower
magnification micrographs near 100x magnification achieve resolutions of 500-1000 nm, while
moderate magnification images near 2000x have resolutions of 25-50 nm.  Although used
somewhat less frequently for these samples due to limited field of view, higher magnification
images of 5000x or above routinely achieve 10 nm resolution.  At magnification above 30kx,
resolution approaching 2 nm is possible in this configuration, however, this results in a very
small field of view without prominent ice facet features, and appears to alter the ice surface
unless very low beam energies are used.  It is someone easier to achieve sub-5 nm resolution,
crisp focus and high contrast images without deforming the ice surfaces if the samples are cryo-
sputter coated and then imaged in high vacuum. However, this process has not been used
frequently because the cryo-sputtering process appears to obscure the smallest nanoscale surface
roughness patterns, and also complicates the prospects for using EDS to measure composition of
aerosol particles.



### 3. Results: Cirrus Ice Crystal Capture

Particularly with respect to detailed visualization of mesoscale roughness and complexity, the Ice Cryo Encapsulation by Balloon (ICE-Ball) probe demonstrates the capability to dramatically enhance knowledge of fine-scale details of cirrus ice particles. In the four successful collection flights from November 2015-August 2017, small numbers (min. 3, max. 20) of intact ice crystals were recovered and imaged by Cryo-SEM. In Spring 2018, the collection aperture was significantly enlarged, which resulted in collection of thousands of crystals on six successful flights during Spring and Summer 2018-2019. The flight on April 24, 2018 was particularly successful, and provides the focus of the results presented here (Fig. 2, Fig. 3, and Table 1) due to the large number of very well-preserved crystals and the synchronous alignment with well-defined NASA A-Train satellite measurements.

The other successful recoveries also yielded significant data, including some marked differences in the morphology of ice crystals captured from the high-altitude clouds. Example ice particle images for these additional flights are provided in supplemental data, along with a description of the synoptic context. Within this sample set, high thin in-situ cirrus (Fig. 4., Supl. 1-E and 1-G) and ice particles within proximity of convection (Supl. 1-C) tended to be smaller and more compact than examples collected from actively growing warm-advection cloud shields (e.g. Fig. 2, Fig. 3, Table 1, and Supl. 1-D).

### 3.1 Synoptic Atmospheric Context on 4/24/2018

On the morning of April 24th, 2018, a surface low pressure system was moving from the Carolinas toward the Northeastern United States. Warm advection aloft generated a shield of ascending air to the north and east of the low, resulting in the emergence of a large region of cirrus and cirrostratus. At mid-morning over central New Jersey, this cirrus deck extended from a 9.3 km base to a 11.5 km cloud top, with an optical depth near 2.0 (NOAA/CIRA analysis algorithms on GOES-16 data). For much of the morning, a faint 22 degree optical halo was visible from the ground in the filtered sunlight, and is also clearly visible from in-flight video (available in supplemental data). The ICE-Ball system was deployed at 11:05 am from near Bordentown, NJ. The payload ascended at approximately 6 m/s, penetrating the ~2 km thick cirrostratus near Ewing, NJ at 11:45 am. Winds at this altitude were 55 kts from the south, with a cloud base temperature of -40°C and a cloud top temperature of -55°C. Video from the flight payload recorded ice particles impacting ICE-Ball for approximately 7 minutes as the instrument




ascended through the cirrus thickness. While the 22 degree halo was clearly evident, no distinct
circumzenith arc was visible on this flight, which was often observed in video at altitude on other
ICE-Ball cirrus penetrations (for example in the Supplement 2: Flight video montage). The
balloon burst at 14 km altitude, and the payload descended via parachute, landing in
Hillsborough, NJ. Recovery occurred approximately 10 minutes after landing, and the captured
and sealed ice particles were transferred into the Cryo-SEM for imaging at approximately 3:00
pm.

### 3.2 Multiform and Intricate Particle Morphology

Captured ice particles from 4/24/2018 and from other flights show striking morphological
diversity and complexity. Despite a collection mechanism that principally reveals particles from
near the top of a single cirrus layer, an extraordinarily wide variety of habits are apparent from
each single cloud penetration, including particles of nearly every cirrus habit classification that is
already recognized (e.g. from Bailey and Hallett, 2009) and several other discernible geometric
forms that have not been reported elsewhere. Among the most striking features of the particle
images is that every aspect of particle morphology is present in multifarious patterns. Even from
one section of one cirrus cloud, and among recognizable particle habits, major inhomogeneities
are present including wide ranges of particle size, aspect ratio, varying degrees of hollowing,
trigonal to hexagonal cross symmetry, broad variations in polycrystallinity, and particles that
range from highly sublimated to those with pristinely sharp edges and facets. Perhaps the best
way to appreciate this immense diversity in particle form is through the stitched mosaic
micrograph from 4/24/18 (Fig. 2). This mosaic of 50 lower-magnification Cryo-SEM images
(100x) captures the entirety of one ICE-Ball sample collection cryo-cell, with a circular inside
diameter of 7.0 mm. Each individual image field is 0.97mm tall x 1.27 mm wide, with a pixel
resolution of 992 nm. An automated multi-capture algorithm on the Hitachi SEM drove the
sample stage to consistent overlap with a high-quality reconstruction; only in the bottom left of
the mosaic is some minor mismatch apparent. The mosaic figure uses false-color to highlight
several particle habits (bullet rosettes, columns, and plates) that fit classic definitions of
morphology. In total, these distinct-habit particles number ~185 of the approximately 1600
individual ice particles that are distinguishable within the depth of focus visible from the top of
the sample. The remaining ~88% of ice particles resolved in figure 2 include the following: a)
complex polycrystal assemblages, often not radiating outward from a single point (~75%), b)





highly sublimated particles where the original habit is no longer distinct (~5% ), c) single bullets
apparently broken off from rosettes (~5%), and d) compact particles with convoluted facets
(~1%). Comparable convoluted crystal forms do not appear to have been reported in the
literature and these particles are labeled as "outre polyhedra". Measurements of cross section
area, ellipse-fit dimensions, solidity, and aspect ratio for these particles are provided in Table 1.
In this sample, the top focal plane reveals only the first several layers of collected crystals. The
full sample collection was accumulated 4 mm deep with an estimated ~35 particle/mm packing,
and thus estimates that approximately 200,000 individual cirrus ice particles were captured and
preserved in this sample alone.
**3.3 Surface Texture Roughness with Multiple Scales and Patterning**
Higher resolution images reveal the topography and textures of crystal facets and edges in
greater detail. Even in the most pristinely faceted crystals that show no evidence of sublimation,
meso-scale texture on the facet surfaces is nearly always apparent at some scale. On some
particles and facets, the roughening is dramatically apparent, with micron-scale features in depth
and wavelength. On other facets, the roughness is significantly more subtle, with dominant
patterning at scales less than 200 nm. In addition, some particles show roughness at multiple
scales simultaneously. While particle complexity and micron-scale roughness are apparent at
100x, resolving the smaller-scale surface textures requires micrograph resolutions of at least 100
nm and carefully tuned contrast. Figure 3 highlights varying degrees of surface roughness in six-
panel micrographs from April 24th, ranging from 250x to 20000x magnification. Panel A. and B.
show examples of the outre polyhedron designation; panel c. demonstrates the open scrolling
seen on a subset of particle facets. It is straightforward to achieve crisp image focus (both
secondary and backscatter) from magnifications of 10x to 5000x in the Hitachi SU5000 with
Quorum cryo-stage, operating at 10-20 Pa in variable pressure mode with stage temperature near
-160℃. Beyond 5000x magnification, crisp focus in variable pressure mode is harder to achieve,
particularly while balancing with a goal of avoiding high beam currents which can induce slight
in-situ sublimation at higher beam energy, density, and exposure times. Nevertheless, at -160℃
and medium beam density, ice particles have extremely low vapor pressure, and even smaller
vapor pressure gradients, such that they can be imaged for hours without noticeable changes in
shape or surface texture at the nm scale. Particles can even be re-sealed while under cryo-



vacuum and removed from the cryo-stage for short-term storage in low-temperature freezers or
liquid nitrogen immersion.

**3.4 Ice-embedded Aerosols and Particulates**

All ice crystal retrievals (and those that did not capture ice) have also collected numerous aerosol
particulates. Although it has not yet been tractable to measure a large fraction of these scavenged
and embedded particulates several dozen have been measured by Energy Dispersive
Spectroscopy (EDAX-EDS), revealing wide-ranging compositions that include mineral dust,
soot, fly-ash, and confirming previous reports of biogenic aerosol (e.g. Pratt et al. 2009). Figure
4 includes three examples of aerosols collected by ICE-Ball, along with EDS spectra of a fly ash
aerosol (Fig. 4b) and iron-rich aerosol particle (Fig. 4c) adhering in the shallow hollow of a
trigonal single crystal. The particulates are highly variable in size, concentration, and
composition, with particles on the surface of crystals, and many additional particles revealed in
the residual samples left by a post-imaging sublimation process in the SEM. As the complex ice
particles sublimate, the embedded aerosol particulates collapse and coagulate with neighboring
particles, and leave a cohesive collection of mixed aerosol particulates near the center of the
original ice crystal. This sublimation-chemical coagulation process may point to a potentially
important cloud-processing effect that could occur during cirrus particle sublimation, possibly
enhancing the ice-nucleating efficiency of the original particulates (Mahrt et al. 2019). As
aerosols of different origin and chemistry conjoin in close proximity under intense sunlight, the
post-sublimation ice particle residuals may serve as an unexpected chemical mixing-pot, altering
the course of their impact on subsequent cirrus formation. Ice particle residuals have been
captured during several previous aircraft field campaigns, but these techniques are primarily
restricted to small ice particles (less than 75 μm) and typically can not provide morphological
imaging of aerosol (Czico and Froyd 2014). With additional flights and increased sampling
statistics, the ICE-Ball aerosol collection technique promises to provide an important
complement to research on the origin and processing pathways of particulates in cirrus clouds
within the high troposphere and across the tropopause.

**4. Conclusions**

Perhaps unsurprisingly, this higher-resolution view of the ice particle constituents of cirrus
reveal new and unanticipated complexities compared to existing laboratory, aircraft, and satellite



measurements. The measurements from ICE-Ball do not contradict laboratory measurements
(Bailey and Hallett 2004) nor do they really dispute the first-order habit diagrams that encompass
cirrus temperatures (Bailey and Hallet 2010). Many of the recent particle observations based on
in-situ imaging from aircraft field campaigns and analysis are also largely corroborated (e.g.
Fridlind et al. 2016, van Diedenhoven et al. 2016a and 2016b, and Lawson et al. 2019).
Nevertheless, present results heighten the appreciation of cirrus particle complexity in four broad
categories:
**4.1  Immense whole-particle habit heterogeneity within single cirrus clouds**
In all cases where multiple crystals were recovered, we observe that the synoptically-forced
cirrus clouds contain a multiplicity of recognizable habit types, even within the same region of
the cloud, and often existing outside of their expected habit temperature and pressure regime. In
addition (and in concurrence with Fridlind et al. 2016 and Lawson et al. 2019), we also find that
a high fraction of particles could be classified as "irregular", in that they do not fit within an
established habit category. The high-resolution images demonstrate that these non-categorized
particles are mainly divided between a) highly-sublimated forms where the original habit is no
longer recognizable and especially b) sharp-edged, faceted particles with complex
polycrystalline morphology that does not neatly fit in established habit categories. In most
instances, these polycrystalline assemblages can not truly be described as plate rosettes, because
the multiple crystals often do not radiate outward from a single focus, and they also frequently
contain plate-like and columnar crystal forms in a single particle. Furthermore, all of the sharp-
edged, and neatly-faceted crystals with no hint of sublimation commonly occur in direct
intermixture with highly sublimated particles. Due to our inherent sampling bias, this observation
may only be particularly apparent at the upper edges of cirrus clouds where entrainment mixing
is prevalent. Nevertheless, this upper-edge region is also of particular radiative importance,
especially in optically thicker cirrus. Despite this diverse morphology within each single cloud,
the set of 9 flight samples also reveals significant patterns of particle variations that appear to be
linked to the dynamical and air-mass characteristics of the cloud. For example, degree of aerosol
loading, average particle size, mean aspect ratio, in-cloud particle concentration, and degree of
polycrystallinity are fairly consistent within each single collection. On one flight, several sets of
collected particles appear as an aggregated chain (Fig. 5c; Supplement 1G.); this cirrus cloud was



not near active convection, but the frontal cirrus original may have derived from modest
convection several hours prior to collection.

### 4.2 Widespread non-hexagonal faceting, hollowing, and scrolls

In addition to unexpectedly convoluted whole-particles, captured ice particle sub-structures and
facets also show a sizeable fraction of trigonal (e.g. Fig. 3b, 4b, 5a), rhomboid (Fig. 3b) and
other non-hexagonal symmetries. In fact, facets with hexagonal symmetry appear to be a slight
minority. For example, columnar single-crystals in figure 2 are shaded in yellow for trigonal or
other-shaped basal cross-sections (53) and green for hexagonal basal cross-section (30). Bullet
rosette cross sections also appear to follow similar proportions. For both bullet rosettes and
columnar habits in figure 2, approximately 80% of crystals demonstrate some degree of
hollowing. This proportion is similar, though slightly higher than reported by Schmitt and
Heymsfield (2007). Smith et al. (2015) also report experiments on the single-scattering impacts
of column hollowing, pointing out that greater hollowing extent tends to increase the asymmetry
parameter, but that the topographical character of the hollowing itself is also important. In
addition to typical center-hollowing, a small fraction (~1%) of ice particles from multiple flights
have prominent "scrolled" geometry (purple in Fig. 2, Fig. 5d) which has been reported in lab
experiments but rarely observed in the atmosphere. Figure 5b shows a set of fairly compact and
relatively small crystals; their unusual convoluted faceting would likely not be recognizable
without resolutions of 1 μm or less. A recent paper by Nelson and Swanson (2019) combine lab
growth experiments with adjoining-surface molecular transport kinetics to explain the
development of "protruding growth" features at laterally-growing ice facets that may be
important contributions to these secondary morphological features. This proposed mechanism
also highlights the role of growth and sublimation cycling in these formations, and helps to
explain the origins of terracing, sheaths, pockets, and trigonal growth, all of which are frequently
observed in ICE-Ball samples.

### 4.3 Mesoscopic roughening at multiple scales and diverse texturing

In high-magnification micrographs with resolution finer that approximately 200 nm, mesoscale
surface roughening on crystal facets and non-faceted sublimation surfaces is nearly always
apparent, but does not appear to occur at a characteristic scale-size or texture pattern in
individual clouds, or even on a single particle. With the smoothest, flattest facets, roughening
patterns may only become apparent with resolutions near or better than 200 nm combined with



carefully tuned contrast. In these instances, the smoothest facets show only subtle topographic
variations with amplitudes smaller than the wavelength of visible light (nano-scale roughening).
Many facets show roughness scales (amplitude and pattern wavelength) on the order of 500 μm
(mesoscopic roughening) and yet others reveal more dramatic roughening with scales in excess
of 1 μm (microscopic scale roughening). In our sample retrieval from 4/24/2018, particles in the
mesoscopic roughening scale range appeared most commonly. We observe that these natural
cirrus particles typically (but not universally) present linear roughening on prism facets and
radial, dendritic, or disordered roughening patterns on basal facets (Fig. 3 panels, and
Supplement 1). These observations of roughening are quite similar to those observed for ice
particles grown within environmental SEM (Magee et al. 2014; Pfalzgraff et al. 2010; Neshyba
et al. 2013, Butterfield et al. 2017) as well a new experimental growth chamber built specifically
to investigate ice surface roughening (Voigtländer et al. 2018). These observations of roughness
at amplitudes and patterning agree with in-situ reports of multi-scale roughness by Collier et al.
2016. The marked similarities in roughness seen on ICE-Ball samples and lab-grown samples
substantiates ESEM and other growth chamber methods as important tools for understanding
mesoscale roughening patterns in cirrus ice growth and sublimation, especially given their
unique ability to observe facets dynamically as they experience growth and sublimation cycling.
**4.4   Composition and morphology of embedded and nucleating aerosol**
Cirrus particles also show high variability with respect to the presence of aerosol particulates
adhered to the crystal surfaces, and embedded within the sub-surface. In the cleanest cases, most
ice particles revealed no obvious (>50 nm radius) non-ice aerosols on the surface (e.g. Fig. 3a),
while in the dirtiest cases (Fig. 4a.; Supplement 1D.), each ice particle averaged several dozen
mineral or pollutant aerosols. Biogenic particulates are also seen with some frequency (Fig. 5e).
While the presence of diverse, rough, complex crystals was striking in every sample collection,
the degree of particulate contamination was highly correlated among individual sample
collections, suggesting that airmass-effects play a dominant role in widely-varying degrees of
aerosol loading. The opportunity to directly image aerosol particle morphology, relationship to
the ice particle surface, and measurement of composition may help to strengthen understanding
of connections between aerosol particles, ice nuclei, ice particle growth, and macro-scale cirrus
properties.





**Figure 1.** ICE-Ball balloon and payload photo at pre-launch (a), with co-authors Lynn, Tusay and Zhao
(left to right). Diagram of servo-driven sealing of cryo-capture vessels and positioning within the ICE-
Ball payload (b).

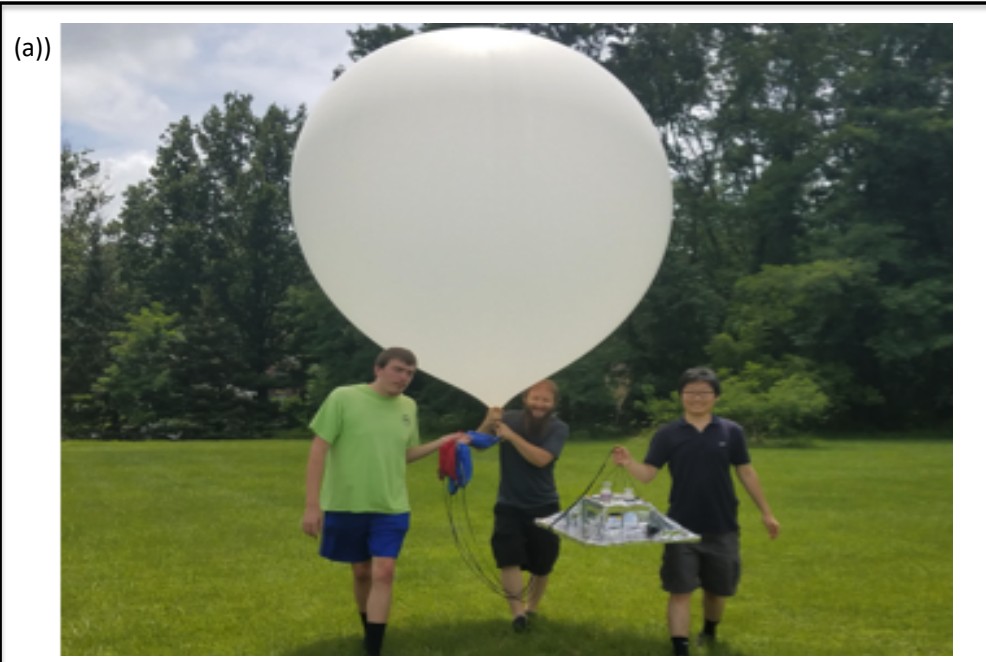

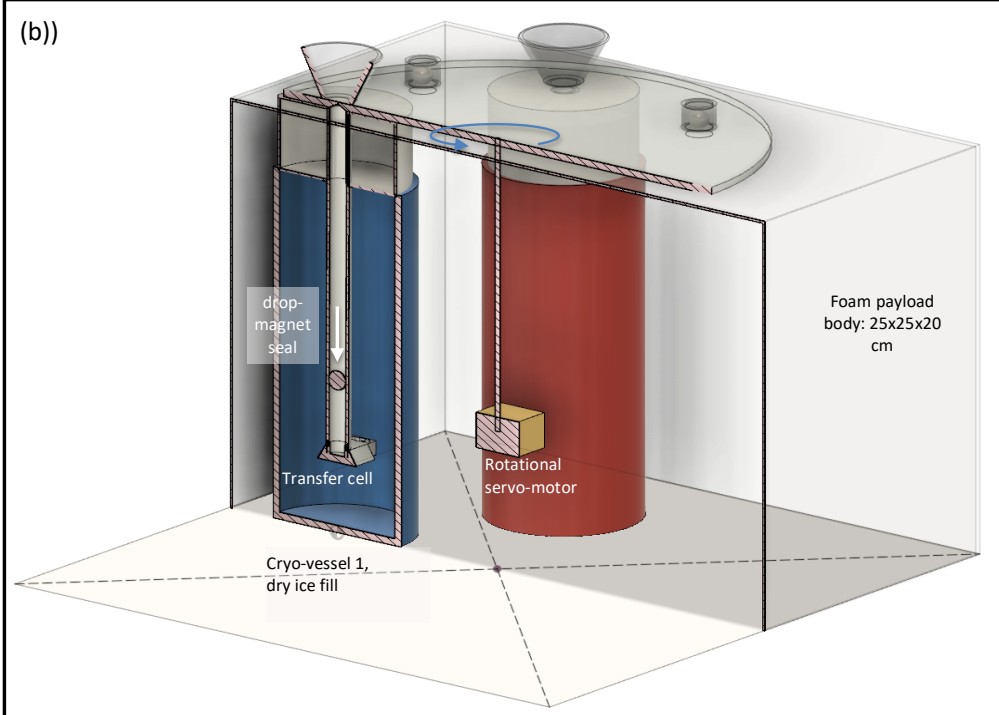





**Figure 2**. Mosaic of 50 Cryo-SEM micrographs of cirrus ice particles captured on 4/24/2018 from ~11
km altitude, -50°C. Each micrograph in this group was acquired at 100x magnification, with resolution of
~900 nm. Actual large circle diameter 7 mm. False color shading groups similar crystal habits, or highly
sublimated particles (orange). Grey-scale particles are sharply-faceted crystals that do not easily fit in
habit classification categories. Table 1 provides class counts and geometric measures.






**Figure 3.** Moderate magnifications (250x to 4500x) of particles, highlighting a wide variety of surface
roughening characteristics. (a) example of a compact-convoluted "outre polyhedron" near several bullet
rosettes and non-classified sharp-faceted particles. On close inspection, multiple patterns of roughness
visible and several mineral aerosols (bright white). (b) rhomboid column with prismatic linear roughening
speckled with discrete surface adhesions, possibly from multiple growth cycles. (c) Rosette with mixed
aspect crystals and an array of geometric surface pits and high mesocopic roughening. (d) Geometrically
tiered and hollowed column of irregular basal cross-section with high roughening. (e) Outre polyhedron
with central hole and irregular roughening. F. High magnification of small, uniform angular roughening.






**Figure 4.** Three-panel Particle images and Energy Dispersive X-ray Spectroscopy (EDAX Octane)
statistics on ice particle contaminants. (a) 100x image of high-aerosol loading on 6/13/2018 (Supplement
1D. for additional details). (b) Fly Ash particle (not ice) captured outside cirrus cloud, with EDS
composition. (c) Shallow hollowed trigonal ice particle with iron-rich embedded aerosol (6/25/2019).




**Figure 5.** Ice particles with non-classical facet features. A. Trigonal crystal with 3 six-sided etch pits,
moderate roughening and aerosol loading. B. Relatively small, compact ice crystals (mean diameter ~55
μm) with convoluted hollowing patterns and moderate mineral dust aerosol load. C. Curving chain of a
~15 particle aggregate including rosettes, compact crystals, and outre polyhedra. D. Moderately
roughened, scrolled plate with corner fins. E. Complex rosette with twisted biogenic particle (left side). F.
Flattened, patterned hexagon with many small adhered aerosols and outre polyhedron (below).






**Table 1.** Statistics for particle habit categories in Figure 2.

| Particle Type | Fig 1. Color | Count | Fit ellipse semi-major mean, µm median | Fit ellipse semi-minor mean, µm median | X-section area mean, µm² median Tot. area% | Aspect ratio mean median | Solidity ratio mean median | Note |
|---|---|---|---|---|---|---|---|---|
| **Columns** | Green and Yellow | 83 | 206 167 | 90 105 | 18200 10900 4.0% | 2.39 2.29 | .88 .90 | Green columns with hexagonal cross section, yellow non-hexagonal. 90% show hollowing |
| **Bullet Rosettes** | Blue | 81 | 189 101 | 90 57 | 18800 13400 4.0% | 1.84 1.60 | .69 .70 | Mean of 6 visible bullets per rosette. Bullets range from thick to very thin and solid to hollow. |
| **Highly sublimated** | Orange | 62 | 139 93 | 77 50 | 18700 3640 3.1% | 1.86 1.69 | .82 .85 | Sublimated to extent original habit and facet shapes not distinguishable. |
| **Plates** | Red and Pink | 20 | 218 204 | 142 125 | 29300 23800 1.5% | 1.64 1.56 | .88 .91 | Red plates nearly hexagonal; pink are non-hexagonal. |
| **Open Scrolls** | Purple | 11 | 183 165 | 124 80 | 21100 17900 0.6% | 1.51 1.53 | .89 .89 | Scroll features overlap with other habits; these show dominant scroll features |
| **Outre Polyhedra** | Teal | 6 | 250 230 | 214 193 | 43000 34200 0.7% | 1.17 1.14 | .88 .91 | Compact particles with convoluted intersecting facets |
| **Complex polycrystals and broken bullets** | Gray | ~13 00 | not measured | not measured | ~86% | not measured | not measured | Sharp-faceted polycrystal particles, often of mixed aspect ratio, including broken bullets (10%) |






**Supplementary Files**
1. Particle images from additional flights (7-slides)
2. Flight video from 4/18/2018 and flight montage
3. Concurrent map, satellite, and meteorological data for 4/24/2018 (4 slides)

**Author Contributions**: NM led ICE-Ball development and deployment. KB and SS made major contributions to system development and data analysis. XZ and EK worked extensively on manuscript figures, supplements, and editing. All co-authors participated in multiple field-campaign flight operations, particle acquisition, instrument engineering, and cryo-imaging.

**Competing interests:** The authors hereby attest they have no competing interests.

**Acknowledgements**: This work was supported in large part by NSF award 1501096, the TCNJ School of Science and Department of Physics, and student support through NSF award 1557357. The Cryo-SEM facility at TCNJ was made possible by the NJ Building our Future Bond Act. The authors thank TCNJ lab manager Rich Fiorillo for many technical contributions and the Allentown FAA field office for gracious support of balloon launches.



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
