# Peer review of "Captured Cirrus Ice Particles in High Definition"

_Atmospheric Chemistry and Physics, 2020_

## Referee Comment (RC1) · Anonymous Referee #1 · 19 Aug 2020

Overall recommendation This study reports not yet revealed and striking detailed morphologies of in-situ ice crystals in natural ice clouds by using the state-of-the-art technology (i.e., cryo-SEM) and somewhat classical balloon capture system (i.e., ICE-Ball). I enjoyed this manuscript and am sure that this study and expected following studies will help to advance our knowledge on complex and not well determined microphysical and radiative properties of individual ice crystals and ice clouds and hence their roles in Earth radiative budget. The overall quality of this manuscript satisfies the standard of the Atmospheric Chemistry and Physics and methods that were used in this study are solid ("seeing is believing"). I recommend this manuscript will be published on Atmospheric Chemistry and Physics with few minor corrections and answering questions listed below.

[Figure]

1. I feel that the result and hence the analysis of this study are somewhat descriptive. More quantitative analysis is required in the following studies. If possible, could the authors add insight or any suggestion on how we can treat (or quantify) the vast variety of complex morphology of natural ice crystals shown in this study to improve parameterization in numerical models and retrieval algorithms in remote sensing?

2. I think that with the current methodology used in the ICE-Ball system it is hard to distinguish whether aerosol particles adhered to crystal surfaces in nature or ice crystals and aerosol particles were sampled separately in nature and then adhered within the collection tube. Can the authors make a comment on this? Do you have a plan to improve the device?

3. The authors need to define "microscale" and "mesoscale".

4. Rework on references is necessary. E.g., van Diedenhoven et al. (2016a) should be Frindlind et al. (2016) in the References section.

5. Page 14, lines 393-194 I think that microscopic and mesoscopic scales are reversed, and "500 um" should be "500 nm".

6. Page 3, line 58 The optical resolution of CPI is 2.3 um.

7. Page 4, line 101 It is "2014-2018", while it is "2016-2019" in the abstract.

8. Page 4, line 105 Please delete "authors Tussay, Lynn, and Zhao holding". It is unnecessary and it is already stated in the caption of Fig. 1.

9. Units should be SI units In this manuscript, non-SI units (e.g., ft and kt) were used.

10. Page 7, line 181 I think that "∼160 C" should be "-160 C".

11. Page 8, line 221 and Supplement 1.C (a) Figure S1.C.(a) is an obviously frozen droplet that is a dominant ice crystal generated by a homogeneous freezing process in the top portion of convective origin ice clouds. This figure is very valuable for the studies on the frozen droplet, frozen droplet aggregates, and homogeneous freezing.

This manuscript will be strengthened by adding the following references:

Stith, J. L., Basarab, B., Rutledge, S. A., and Weinheimer, A.: Anvil microphysical signatures associated with lightning- produced NOx, Atmos. Chem. Phys., 16, 2243–2254, https://doi.org/10.5194/acp-16-2243-2016, 2016.

Um, J., G. M. McFarquhar, J. L. Stith, C. H. Jung, S. S. Lee, J. Y. Lee, Y. Shin, Y. G. Lee, Y. I. Yang, S. S. Yum, B.-G. Kim, J. W. Cha, and A.-R. Ko, 2018: Microphysical characteristics of frozen droplet aggregates from deep convective clouds. Atmos. Chem. Phys., 18, 16915-16930, https://doi.org/10.5194/acp-18-16915-2018.

12. Page 9, lines 262-264 Can the authors add the explanation of the habit classification method used here? Is it manual identification?

13. Page 10, line 272 Can the authors define "solidity" here or "solidity ratio" in Table 1?

14. Page 10, line 282 What does "wavelength" mean here?

15. Page 10, line 288 "panel c" -> "panel d" in Fig. 3. In Fig. 3, panel labels "(d)" and "(e)" should be exchanged.

16. Page 10, line 290 "secondary and backscatter" -> "secondary and backscattered electrons" would be clearer.

17. Page 12, line 346 ". . .. as plate rosettes, . . .." Is it ". . .. as bullet rosettes"?

18. Page 13, line 363 Fig. 4b is an ash particle.

19. Page 17, Fig. 3 I think that the panel labels "(d)" and "(e)" should be exchanged. "F" should be "(f)" in the figure caption.

20. Supplement 1.D The authors need to add panel labels. The "panel b" is called in the caption.

21. Supplement 1.D, caption Please add "(f)" after ". . .. Complex mineral aerosol

particles".

---

## Referee Comment (RC2) · Anonymous Referee #2 · 20 Aug 2020

**1  Content**

The manuscript is about ice crystal observations with a new balloon borne instrument device. With this instrument the ice crystals are captured during the flight, conserved and analysed with a scanning electron microscope (SEM) in the laboratory. With this technique they found a larger variety of ice crystals shapes and geometries as well as surface roughness.

[Figure]

**2 Overall impression and rating**

The overall impression of the manuscript is good. The manuscript is mostly easy to understand and to my opinion enough structured. This novel technique of capturing ice crystals and detailed analysis of their surface will enhance the knowledge of which types of crystals and their fine structure can be found in the atmosphere. I really like the detailed SEM pictures of ice crystals and your video is also nice to watch. I agree mostly with the interpretations and I think that the manuscript is a good contribution to the science community. I have some smaller concerns, which should be addressed before publication. For these reasons, I recommend publication in ACP after minor revisions.

**3 Specific comments/questions:**

- **Sampling characteristic**
  The focus of the manuscript is more on the results of the different balloon soundings, which is of course important to be a publication in ACP. However, I think that there should be a bit more technical information about the sampling characteristic of the instrument, which is important to understand the observations. For example you mentioned that the efficiency of collection is high for particles larger than 50 microns. In Luebke et al. 2016 Figure 10 you find averaged cirrus size distributions of different cloud types which show that a large fraction of ice crystals are also below 50 microns in diameter. If those particles would not enter your sample device you would get only the large crystals which would lead to a distortion of your cloud statistic. Therefore, it would be good if you can provide more information like lower/upper cutoff size, sampling efficiency for different particle sizes, sampling volume, minimum number concentration in Section 2.1.

- **Mapping of microphysical properties to atmospheric conditions**
  As far as I understood the sampling device just samples the crystals from the bottom to the top of a cloud consecutively in one or maximum two sample probes. In case that the number concentration in the cloud is rather low, I can imaging that the mapping of ice crystals found in the sampler to the location and thus to the atmospheric condition (temperature, pressure, humidity) is not really possible. You always find a mixture of particles form the whole cloud column. The other extreme would be a very high number concentration of crystals in the cloud. In this case you see so many crystals on top of each other that you only can see the cloud top in the upper layer of your probe. Than you do not have a full picture of the whole cloud. With this two examples I cannot fully follow the argument of Section 4.1, where you stated to find a large habit heterogeneity within single clouds. You should discuss this point in more detail and maybe also assess which impact do you see your statement of this section due to the sampling.

- **Sampling of different cirrus cloud types**
  At some point in the text (best in Section 2.3) you should mentioned that you focus mostly on thick cirrus layer as they occur typically within frontal systems like warm convenor belts. This is mainly caused by your launch planning/preparation approach and the better predictability of such frontal cirrus clouds. These clouds have typically a large ascent (see e.g. your trajectory with ascent from 5.5 to 11 km in the supplementary material) bringing high amount of moisture into the cirrus altitude. These clouds typically pass through the mixed phase temperature range above -38°C and are referred in the literature as liquid origin cirrus clouds (e.g. Luebke et al. 2016 or Wernli et al. 2016). Ice crystals in these clouds are typically larger in size and show a more complex shape compared to in-situ formed cloud at cirrus altitudes. I suggest to mention these in your text that your results are mostly representative for liquid origin clouds and may not be meaningful for in-situ formed clouds.

- Page 8, lines 220-223: The influence of cloud origin and dynamics on crystal size is in agreement with other studies. Here, I would recommend to cite the paper by Luebke et al. 2016.

- Page 12, lines 351-352: I cannot follow your argumentation that you found sublimated ice crystals at the cloud top due to entrainment. Usually, the cloud top is dominated by nucleation of crystals and there you have the coldest temperatures and highest relative humidity wrt. ice (see e.g. Spichtinger and Gierens 2009). Thus, to find sublimated ice crystals at cloud top seems to unrealistic. At least this argument needs more explanation, citations etc and also discussion with point above.

- Figure 2: What is the large "rock" in the upper left part of the SEM picture. Maybe you can mentioned this also in the text because it is very conspicuous.

**4 Technical comments/suggestions:**

- **Units in the manuscript**
  In ACP usually all values and their units are given as SI base unit. For example you use the kt for the wind speed which should be given in meters per second (m/s). This unit is also recommended by the World Meteorological Organization for reporting wind speeds. I would recommend to go through the entire manuscript and change all non SI units like miles etc. to appropriate SI unit.

- Page 2, line 43: I suggest to cite also Sourdeval et al. 2018 to have one representative paper of cirrus properties using lidar/radar technique.

- Page 7, line 178: "Hitachi SU5000 is employs a Schottky ", the word "is" is to much.

- Page 7, line 181: Minus is missing at the temperature value. Should be ~-160°C

- Page 10, line 288: Capitalize the "c" --> panel C.

- Page 12, line 334-335: Please use another word than categories, because the reader expect than particles to be sorted in specific categories which is not the case here. It is more like a list of all the findings. You should use e.g. topics or findings.

- Page 13, line 385: "finer than" instead of "finder that"

- Page 14, line 392: I think you mean 500nm instead of 500 microns

- Figure 4: a) No scale, please add a scale here. b)-left and c)-left Scale not readable. b)-right and c)-right Table not readable, please enlarge or skip. Peak classification in the diagramm not readable, please enlarge Peak labels.

- Figure Supl. 1-F (a) and 1-G (a-f): Scale not readable. Can please add the same gray shadow behind the scale as you did in the other pictures.

**5 References:**

- Luebke, A. E., Afchine, A., Costa, A., Grooß, J.-U., Meyer, J., Rolf, C., Spelten, N., Avallone, L. M., Baumgardner, D., and Krämer, M.: The origin of midlatitude ice clouds and the resulting influence on their microphysical properties, Atmos. Chem. Phys., 16, 5793–5809, https://doi.org/10.5194/acp-16-5793-2016, 2016.

- Sourdeval, O., Gryspeerdt, E., Krämer, M., Goren, T., Delanoë, J., Afchine, A., Hemmer, F., and Quaas, J.: Ice crystal number concentration estimates from

lidar–radar satellite remote sensing – Part 1: Method and evaluation, Atmospheric Chemistry and Physics, 18, 14 327–14 350, doi:10.5194/acp-18-14327-2018, 2018.

- Spichtinger, P. and Gierens, K. M.: Modelling of cirrus clouds – Part 1b: Structuring cirrus clouds by dynamics, Atmos. Chem. Phys., 9, 707–719, https://doi.org/10.5194/acp-9-707-2009, 2009.

- Wernli, H., Boettcher, M., Joos, H., Miltenberger, A. K., and Spichtinger, P. (2016), A trajectory‐based classification of ERA‐Interim ice clouds in the region of the North Atlantic storm track, Geophys. Res. Lett., 43, 6657– 6664, doi:10.1002/2016GL068922.

---

## Author Response (AR1)

Overall recommendation This study reports not yet revealed and striking detailed morphologies of in-situ ice crystals in natural ice clouds by using the state-of-the-art technology (i.e., cryo-SEM) and somewhat classical balloon capture system (i.e., ICE-Ball). I enjoyed this manuscript and am sure that this study and expected following studies will help to advance our knowledge on complex and not well determined microphysical and radiative properties of individual ice crystals and ice clouds and hence their roles in Earth radiative budget. The overall quality of this manuscript satisfies the standard of the Atmospheric Chemistry and Physics and methods that were used in this study are solid ("seeing is believing"). I recommend this manuscript will be published on At- mospheric Chemistry and Physics with few minor corrections and answering questions listed below.

1.  I feel that the result and hence the analysis of this study are somewhat descrip- tive. More quantitative analysis is required in the following studies. If possible, could the authors add insight or any suggestion on how we can treat (or quantify) the vast variety of complex morphology of natural ice crystals shown in this study to improve parameterization in numerical models and retrieval algorithms in remote sensing?

2.  I think that with the current methodology used in the ICE-Ball system it is hard to distinguish whether aerosol particles adhered to crystal surfaces in nature or ice crystals and aerosol particles were sampled separately in nature and then adhered within the collection tube. Can the authors make a comment on this? Do you have a plan to improve the device?

3.  The authors need to define "microscale" and "mesoscale".

4.  Rework on references is necessary. E.g., van Diedenhoven et al. (2016a) should be Frindlind et al. (2016) in the References section.

5.  Page 14, lines 393-194 I think that microscopic and mesoscopic scales are reversed, and "500 um" should be "500 nm".

6.  Page 3, line 58 The optical resolution of CPI is 2.3 um.

7.  Page 4, line 101 It is "2014-2018", while it is "2016-2019" in the abstract.

8.  Page 4, line 105 Please delete "authors Tussay, Lynn, and Zhao holding". It is unnecessary and it is already stated in the caption of Fig. 1.

9.  Units should be SI units In this manuscript, non-SI units (e.g., ft and kt) were used.

10. Page 7, line 181 I think that "~160 C" should be "-160 C".

11. Page 8, line 221 and Supplement 1.C (a) Figure S1.C.(a) is an obviously frozen droplet that is a dominant ice crystal generated by a homogeneous freezing process in the top portion of convective origin ice clouds. This figure is very valuable for the studies on the frozen droplet, frozen droplet aggregates, and homogeneous freezing.

This manuscript will be strengthened by adding the following references:

Stith, J. L., Basarab, B., Rutledge, S. A., and Weinheimer, A.: Anvil microphysical signatures associated with lightning- produced NOx, Atmos. Chem. Phys., 16, 2243–2254, https://doi.org/10.5194/acp-16-2243-2016, 2016.

Um, J., G. M. McFarquhar, J. L. Stith, C. H. Jung, S. S. Lee, J. Y. Lee, Y. Shin, Y. G. Lee, Y. I. Yang, S. S. Yum, B.-G. Kim, J. W. Cha, and A.-R. Ko, 2018: Microphysical characteristics of frozen droplet aggregates from deep convective clouds. Atmos. Chem. Phys., 18, 16915-16930, https://doi.org/10.5194/acp-18-16915-2018.

12. Page 9, lines 262-264 Can the authors add the explanation of the habit classifica- tion method used here? Is it manual identification?

13. Page 10, line 272 Can the authors define "solidity" here or "solidity ratio" in Table 1?

14. Page 10, line 282 What does "wavelength" mean here?

15. Page 10, line 288 "panel c" -> "panel d" in Fig. 3. In Fig. 3, panel labels "(d)" and "(e)" should be exchanged.

16. Page 10, line 290 "secondary and backscatter" -> "secondary and backscattered electrons" would be clearer.

17. Page 12, line 346 "…. as plate rosettes, …." Is it "…. as bullet rosettes"?

18. Page 13, line 363 Fig. 4b is an ash particle.

19. Page 17, Fig. 3 I think that the panel labels "(d)" and "(e)" should be exchanged. "F" should be "(f)" in the figure caption.

20. Supplement 1.D The authors need to add panel labels. The "panel b" is called in the caption.

21. Supplement 1.D, caption Please add "(f)" after ". . .. Complex mineral aerosol particles".
The manuscript is about ice crystal observations with a new balloon borne instrument device. With this instrument the ice crystals are captured during the flight, conserved and analysed with a scanning electron microscope (SEM) in the laboratory. With this technique they found a larger variety of ice crystals shapes and geometries as well as surface roughness.

**2 Overall impression and rating**

The overall impression of the manuscript is good. The manuscript is mostly easy to understand and to my opinion enough structured. This novel technique of capturing ice crystals and detailed analysis of their surface will enhance the knowledge of which types of crystals and their fine structure can be found in the atmosphere. I really like the detailed SEM pictures of ice crystals and your video is also nice to watch. I agree mostly with the interpretations and I think that the manuscript is a good contribution to the science community. I have some smaller concerns, which should be addressed before publication. For these reasons, I recommend publication in ACP after minor revisions.

**3 Specific comments/questions:**

• **Sampling characteristic**

The focus of the manuscript is more on the results of the different balloon sound- ings, which is of course important to be a publication in ACP. However, I think that there should be a bit more technical information about the sampling charac- teristic of the instrument, which is important to understand the observations. For example you mentioned that the efficiency of collection is high for particles larger than 50 microns. In Luebke et al. 2016 Figure 10 you find averaged cirrus size distributions of different cloud types which show that a large fraction of ice crys- tals are also below 50 microns in diameter. If those particles would not enter your sample device you would get only the large crystals which would lead to a distor- tion of your cloud statistic. Therefore, it would be good if you can provide more information like lower/upper cutoff size, sampling efficiency for different particle sizes, sampling volume, minimum number concentration in Section 2.1

• **Mapping of microphysical properties to atmospheric conditions**

As far as I understood the sampling device just samples the crystals from the bottom to the top of a cloud consecutively in one or maximum two sample probes. In case that the number concentration in the cloud is rather low, I can imaging that the mapping of ice crystals found in the sampler to the location and thus to the atmospheric condition (temperature, pressure, humidity) is not really possible. You always find a mixture of particles form the whole cloud column. The other extreme would be a very high number concentration of crystals in the cloud. In this case you see so many crystals on top of each other that you only can see the cloud top in the upper layer of your probe. Than you do not have a full picture of the whole cloud. With this two examples I cannot fully follow the argument of Section 4.1, where you stated to find a large habit heterogeneity within single clouds. You should discuss this point in more detail and maybe also assess which impact do you see your statement of this section due to the sampling.

- **Sampling of different cirrus cloud types**

At some point in the text (best in Section 2.3) you should mentioned that you focus mostly on thick cirrus layer as they occur typically within frontal systems like warm convenor belts. This is mainly caused by your launch planning/preparation approach and the better predictability of such frontal cirrus clouds. These clouds have typically a large ascent (see e.g. your trajectory with ascent from 5.5 to 11 km in the supplementary material) bringing high amount of moisture into the cirrus altitude. These clouds typically pass through the mixed phase temperature range above -38°C and are referred in the literature as liquid origin cirrus clouds (e.g. Luebke et al. 2016 or Wernli et al. 2016). Ice crystals in these clouds are typically larger in size and show a more complex shape compared to in-situ formed cloud at cirrus altitudes. I suggest to mention these in your text that your results are mostly representative for liquid origin clouds and may not be meaningful for in-situ formed clouds.

- Page 8, lines 220-223: The influence of cloud origin and dynamics on crystal size is in agreement with other studies. Here, I would recommend to cite the paper by Luebke et al. 2016.

- Page 12, lines 351-352: I cannot follow your argumentation that you found sub- limated ice crystals at the cloud top due to entrainment. Usually, the cloud top is dominated by nucleation of crystals and there you have the coldest tempera- tures and highest relative humidity wrt. ice (see e.g. Spichtinger and Gierens 2009). Thus, to find sublimated ice crystals at cloud top seems to unrealistic. At least this argument needs more explanation, citations etc and also discussion with point above.

- Figure 2: What is the large "rock" in the upper left part of the SEM picture. Maybe you can mentioned this also in the text because it is very conspicuous.

**4 Technical comments/suggestions:**

- **Units in the manuscript**
  In ACP usually all values and their units are given as SI base unit. For example you use the kt for the wind speed which should be given in meters per sec- ond (m/s). This unit is also recommended by the World Meteorological Organi- zation for reporting wind speeds. I would recommend to go through the entire manuscript and change all non SI units like miles etc. to appropriate SI unit.

- Page 2, line 43: I suggest to cite also Sourdeval et al. 2018 to have one repre- sentative paper of cirrus properties using lidar/radar technique.

- Page 7, line 178: "Hitachi SU5000 is employs a Schottky ", the word "is" is to much.

-  7, line 181: Minus is missing at the temperature value. Should be ~-160°C

- Page 10, line 288: Capitalize the "c" --> panel C.

- Page 12, line 334-335: Please use another word than categories, because the reader expect than particles to be sorted in specific categories which is not the case here. It is more like a list of all the findings. You should use e.g. topics or findings.

- Page 13, line 385: "finer than" instead of "finder that"

- Page 14, line 392: I think you mean 500nm instead of 500 microns

- Figure 4: a) No scale, please add a scale here. b)-left and c)-left Scale not readable. b)-right and c)-right Table not readable, please enlarge or skip. Peak classification in the diagramm not readable, please enlarge Peak labels.

- Figure Supl. 1-F (a) and 1-G (a-f): Scale not readable. Can please add the same gray shadow behind the scale as you did in the other pictures.

**5 References:**

- Luebke, A. E., Afchine, A., Costa, A., Grooß, J.-U., Meyer, J., Rolf, C., Spelten, N., Avallone, L. M., Baumgardner, D., and Krämer, M.: The origin of midlatitude ice clouds and the resulting influence on their microphysical properties, Atmos. Chem. Phys., 16, 5793–5809, https://doi.org/10.5194/acp-16-5793-2016, 2016.

- Sourdeval, O., Gryspeerdt, E., Krämer, M., Goren, T., Delanoë, J., Afchine, A., Hemmer, F., and Quaas, J.: Ice crystal number concentration estimates from lidar–radar satellite remote sensing – Part 1: Method and evaluation, Atmo- spheric Chemistry and Physics, 18, 14 327–14 350, doi:10.5194/acp-18-14327-2018, 2018.

- Spichtinger, P. and Gierens, K. M.: Modelling of cirrus clouds – Part 1b: Struc- turing cirrus clouds by dynamics, Atmos. Chem. Phys., 9, 707–719, https://doi.org/10.5194/acp-9-707-2009, 2009.

- Wernli, H., Boettcher, M., Joos, H., Miltenberger, A. K., and Spichtinger, P. (2016), A trajectory‐based classification of ERA‐Interim ice clouds in the region of the North Atlantic storm track, Geophys. Res. Lett., 43, 6657– 6664, doi:10.1002/2016GL068922.

**2. Authors' Response**

The authors would like to thank the two referees for your thoughtful and detailed comments on this manuscript. Both reviews bring up important questions and considerations that should help to strengthen the final version of the paper. With respect to specific comments and questions from the referees, we convey our replies as follows:

In reply to RC1:
1. We agree that the results of the present manuscript are mostly descriptive rather than thoroughly quantified. We do think that these cryo-SEM images and the qualitative assessments thereof nevertheless present a novel view of cirrus ice particles that give insight to important questions, and suggest new avenues to pursue regarding cirrus microphysical research. In some senses, the inherently limited sampling characteristics of this technique challenge the creation of quantifiable measures of large-scale cirrus properties, so we were reluctant to offer definitive statistics or parameterizations on a few samples that may not be broadly representative. However, our continuing work is pursuing several avenues to make ICE-Ball results more quantifiable and extendable to parameterizations for modeling: a) we have added a particle-vision system to be able to quantify cirrus particle densities coincident with particle captures, b) we are collaborating on a new project with PSU and DOE-ARM to fly ICE-Ball missions in different synoptic regimes and in conjunction with cloud RADAR and LIDAR remote sensing, and c) new research team members will focus on turning SEM micrographs of the ice particles into quantifiable statistics of several measures of complexity. In the final manuscript, we will add a paragraph to the conclusions highlighting potential paths forward toward improved quantification of cirrus ice particle complexity.

2. We think insights into aerosol roles in cirrus are a potential high-value contribution from ICE-Ball data in future work. We are confident that most of the small visible aerosol particles we see in the micrographs have adhered to the ice crystals in-cloud for several reasons: a) most aerosols are firmly attached or partially embedded in the ice surfaces (we can move and tilt the stage and partially sublimate surfaces to confirm this), b) CFD simulations and clear-sky flights both indicate that our system has a low collection efficiency for particles below 20 micron diameter, and c) collection tubes are normally sealed closed except when near or inside the cirrus clouds. New versions of the ICE-Ball sampling system will further address this question through a redesigned flow-path that aims to improve small-particle collection efficiency, in conjunction with the coincident particle vision system mentioned above. We are also planning for more complete imaging, counting, and compositional characterization of interstitial, surface-embedded, and residual aerosols in future missions.

3. - 21. (Technical comments, added references, and editing suggestions) Thank you very much for providing these detailed suggestions and additional references. All units will be formatted to SI standard notation. We look forward to incorporating each of these additions and corrections in the final manuscript.

In reply to RC2:
1. Sampling Characteristic:

We agree that a more thorough understanding and presentation of sampling characteristics of the ICE-Ball system is important to allow the measurements to be put in proper context relative to a single cirrus cloud, let alone to present measurements as characteristic to all cirrus or subtypes of cirrus. We have CFD simulations of the system geometry and streamlines that broadly appear to agree with the data from in-flight ice and aerosol particle collections. The CFD results and actual data both show collection efficiencies near 100% for particles over 60 micron max. diameter and efficiencies becoming quite small for particles under 20 microns. For example, the flight from April 18, 2019 (supplement 1E) mostly collected particles with diameters between 30-70 microns; most other flight collections were dominated by larger particles, with only a few below 50 microns. The regime between 20-60 micron diameter appears to be a transition where we capture a fraction of particles near the capture-tube inlets, but with higher fractions of the increasingly small particles following streamlines around the inlet tube. We aren't yet confident enough to assign exact collection efficiencies as a function of particle size because the collection characteristics appear to also be influenced by small-scale cloud turbulence, balloon-wake flow, and particle density. However, as discussed in our reply to RC1 above, we are adding an in-flight particle vision system to have an independent means to observe particle numbers, and we are slightly redesigning the capture aerodynamics to better sample smaller particles. Nevertheless, we would be happy to add some of the CFD results for the configuration used in this paper along with a sampling discussion. This could be included in the supplement, or if the editor prefers, we could add a figure and discussion to the methods section of the final version of the paper.

2. Mapping of microphysical properties to atmospheric conditions:
This is a good point, and your description of the system's consecutive sampling of the cloud column is correct. As you described, in the case of a high-density collection of many particles (e.g. figure 2), it is very likely that collections from the bottom of the cloud are buried under the top few layers of particles that we can image, and were captured only from the top of the cirrus layer. It is from several of these situations that we inferred high habit heterogeneity in the tops of these clouds, including the mix of sharp-faceted particles with others having high sublimation. We agree that the more sparse collections don't provide good insight into the distribution of habits through the cloud layer (except in a few cases where particle shapes are not highly heterogenous, e.g. supplement 1E). We do expect that future missions can provide significantly more insight into vertical distribution of ice particle habits and help unravel connections to cloud dynamics and thermodynamics. The planned missions over US Dept. of Energy-ARM RADAR and LIDAR are particularly exciting in this regard. In the case of a future flight through a relatively thick or dense cirrus layer, we will also plan to isolate captures by altitude, to sample the bottom, middle, and top of the cloud layer separately. In any case, we agree it is a good idea to revise and add detail to our current discussion in 4.1, and also point to the planned improvements in future work.

3. Sampling of different cirrus cloud types:
We agree that we should explicitly point out that the focal data set in Fig. 2, Fig. 3, and table 1, and several of the additional data shown in the supplement (1A & 1D) constitute moderately thick frontal cirrus, although in none of the sampled cirrus were thick enough to be optically opaque. Several of the supplement data collections are from thin cirrus (1B,1E,1F,1G) or convective-origin cirrus (1C). We will look forward to including the insights from the papers you recommend in a revised discussion of the atmospheric and cloud-scale context of the cirrus particles described in Fig. 2, Fig. 3, and table 1.  Overall, we very much agree that distinct cirrus types need to sampled more comprehensively, with a goal to connect detailed ice particle characteristics with the full range of cirrus altitudes, temperatures, dynamics, and nucleation modes.

Technical comments, suggestions, and references:
Thank you very much for providing valuable additional references to include, and for several detailed editing corrections.  We look forward to incorporating each of these additions and corrections in the final manuscript.  We will revise to ensure all units are given in the SI standard.  Finally, with regard to the question about the "rock" at upper left in figure 2: we think this is a steel burr (remnant from machining) that broke off from inside the collection tube. No such large mm-scale particles were observed in any other particle captures.

**3.  Changes in Manuscript**

Changes are described in *blue italics*; line numbers refer to the revised manuscript Word document with tracked changes visible (appended below).

**Regarding Referee 1 comments:**

1.  I feel that the result and hence the analysis of this study are somewhat descrip- tive. More quantitative analysis is required in the following studies. If possible, could the authors add insight or any suggestion on how we can treat (or quantify) the vast variety of complex morphology of natural ice crystals shown in this study to improve parameterization in numerical models and retrieval algorithms in remote sensing?

    *As we mentioned in the author response comment above, we do not disagree with this characterization.  A thorough method to capture and parameterize the vast complexity of cirrus microphysics will undoubtably be a big project that require input and consensus-building from many researchers.  We do hope that future ICE-Ball missions will help with this!  We aim to sample a wider array of cirrus clouds and include expanded particle analyses, as well as plans to coordinate with remote sensing and modeling.  We have pointed to several of these goals in additional descriptions in section 2., and we will take this suggestion to heart in our upcoming projects.*

2.  I think that with the current methodology used in the ICE-Ball system it is hard to distinguish whether aerosol particles adhered to crystal surfaces in nature or ice crystals and aerosol particles were sampled separately in nature and then adhered within the collection tube. Can the authors make a comment on this? Do you have a plan to improve the device?

    *Additional clarification on this point was added in the beginning of section 3.4 (revised manuscript lines 340-344), reflecting the author response comment above.*

3.  The authors need to define "microscale" and "mesoscale".

    *This definition has been added to the introduction, revised manuscript lines 51-53.*

4.  Rework on references is necessary. E.g., van Diedenhoven et al. (2016a) should be Frindlind et al. (2016) in the References section.

    *This reference has been corrected, and other references checked for format consistency.*

5.  Page 14, lines 393-194 I think that microscopic and mesoscopic scales are reversed, and "500 um" should be "500 nm".

    *You are correct, this reversal has been corrected (line 435).*

6.  Page 3, line 58 The optical resolution of CPI is 2.3 um.

    *The pixel resolution is 2.3 um, but Lawson et al. 2019 indicate an effective ~5 um optical resolution.  This clarification has been added at line 60.*

7.  Page 4, line 101 It is "2014-2018", while it is "2016-2019" in the abstract.

    *The full date range for flights is actually 2015-2019; this has been corrected in both locations.*

8.  Page 4, line 105 Please delete "authors Tussay, Lynn, and Zhao holding". It is unnecessary and it is already stated in the caption of Fig. 1.

*This has been removed.*

9. Units should be SI units In this manuscript, non-SI units (e.g., ft and kt) were used.

   *All instances of non-SI units have been changed to SI standard units.*

10. Page 7, line 181 I think that "~160 C" should be "-160 C".

    *This typo has been corrected.*

11. Page 8, line 221 and Supplement 1.C (a) Figure S1.C.(a) is an obviously frozen droplet that is a dominant ice crystal generated by a homogeneous freezing process in the top portion of convective origin ice clouds. This figure is very valuable for the studies on the frozen droplet, frozen droplet aggregates, and homogeneous freezing.

This manuscript will be strengthened by adding the following references:

Stith, J. L., Basarab, B., Rutledge, S. A., and Weinheimer, A.: Anvil microphysical signatures associated with lightning- produced NOx, Atmos. Chem. Phys., 16, 2243– 2254, https://doi.org/10.5194/acp-16-2243-2016, 2016.

Um, J., G. M. McFarquhar, J. L. Stith, C. H. Jung, S. S. Lee, J. Y. Lee, Y. Shin, Y. G. Lee, Y. I. Yang, S. S. Yum, B.-G. Kim, J. W. Cha, and A.-R. Ko, 2018: Microphysical characteristics of frozen droplet aggregates from deep convective clouds. Atmos. Chem. Phys., 18, 16915-16930, https://doi.org/10.5194/acp-18-16915-2018.

   *We are pleased to add these two references () and highlight the importance of convective-origin cirrus microphysics, including a high-priority goal to sample anvil cirrus in upcoming ICE-Ball missions (lines 250-255).*

12. Page 9, lines 262-264 Can the authors add the explanation of the habit classifica- tion method used here? Is it manual identification?

    *That is correct, standard habit classifications were done manually, with consensus required among 3 co-authors. This clarification is added at line 301.*

13. Page 10, line 272 Can the authors define "solidity" here or "solidity ratio" in Table 1?

    *The particle measurement method and solidity definition have been added at lines 311-315.*

14. Page 10, line 282 What does "wavelength" mean here?

    *Wavelength refers here to the dimension between ridges of roughening. This has been clarified at line 324.*

15. Page 10, line 288 "panel c" -> "panel d" in Fig. 3. In Fig. 3, panel labels "(d)" and "(e)" should be exchanged.

    *Thank you very much for catching this error – the panel labels have been corrected.*

16. Page 10, line 290 "secondary and backscatter" -> "secondary and backscattered electrons" would be clearer.

    *This clarification has been added.*

17. Page 12, line 346 ".… as plate rosettes, .…" Is it ".… as bullet rosettes"?

    *This sentence is intended to distinguish between complex polycrystal assemblages vs polycrystals "rosettes" that radiate from a common origin point (with the usual bulleted columnar arms or with plate-like arms). This has been clarified at line 393.*

18. Page 13, line 363 Fig. 4b is an ash particle.

*Agreed, this is noted at line 354 and in the figure caption..*

19. Page 17, Fig. 3 I think that the panel labels "(d)" and "(e)" should be exchanged. "F" should be "(f)" in the figure caption.

20. Supplement 1.D The authors need to add panel labels. The "panel b" is called in the caption.

*Thank you very much for catching these errors. All figure panel labels have been double checked and corrected for consistency and accurate referencing in the text.*

21. Supplement 1.D, caption Please add "(f)" after ". . .. Complex mineral aerosol particles".

*This has been added to supplement 1D.*

**Regarding Referee 2 comments, with changes described in italics**

**Sampling characteristic**

The focus of the manuscript is more on the results of the different balloon sound- ings, which is of course important to be a publication in ACP. However, I think that there should be a bit more technical information about the sampling charac- teristic of the instrument, which is important to understand the observations. For example you mentioned that the efficiency of collection is high for particles larger than 50 microns. In Luebke et al. 2016 Figure 10 you find averaged cirrus size distributions of different cloud types which show that a large fraction of ice crys- tals are also below 50 microns in diameter. If those particles would not enter your sample device you would get only the large crystals which would lead to a distor- tion of your cloud statistic. Therefore, it would be good if you can provide more information like lower/upper cutoff size, sampling efficiency for different particle sizes, sampling volume, minimum number concentration in Section 2.1

*We have added text regarding particle collection efficiency at lines 123-126, and we have added a streamline analysis figure from CFD simulations to supplement 1. We have also added a subsection in the methods description for current upgrades to the ICE-Ball instrument that will include particle-vision video to cross-check particle captures with cloud particle density and to allow for separation of samples from different altitudes within a cloud (lines 221-231).*

**Mapping of microphysical properties to atmospheric conditions**

As far as I understood the sampling device just samples the crystals from the bottom to the top of a cloud consecutively in one or maximum two sample probes. In case that the number concentration in the cloud is rather low, I can imaging that the mapping of ice crystals found in the sampler to the location and thus to the atmospheric condition (temperature, pressure, humidity) is not really possible. You always find a mixture of particles form the whole cloud column. The other extreme would be a very high number concentration of crystals in the cloud. In this case you see so many crystals on top of each other that you only can see the cloud top in the upper layer of your probe. Than you do not have a full picture of the whole cloud. With this two examples I cannot fully follow the argument of Section 4.1, where you stated to find a large habit heterogeneity within single clouds. You should discuss this point in more detail and maybe also assess which impact do you see your statement of this section due to the sampling.

*This explanation has been expanded in section 3.2 (lines 279-283) and clarified to point out that the inference that a heterogenous set of particles are only from near cloud top is limited only to dense collections of many particles, as in Figure 2.*

**Sampling of different cirrus cloud types**

At some point in the text (best in Section 2.3) you should mentioned that you focus mostly on thick cirrus layer as they occur typically within frontal systems like warm convenor belts. This is mainly caused by your launch planning/preparation approach and the better predictability of such frontal cirrus clouds. These clouds have typically a large ascent (see e.g. your trajectory with ascent from 5.5 to 11 km in the supplementary material) bringing high amount of moisture into the cirrus altitude. These clouds typically pass through the mixed phase temperature range above -38°C and are referred in the literature as liquid origin cirrus clouds (e.g. Luebke et al. 2016 or Wernli et al. 2016). Ice crystals in these clouds are typically larger in size and show a more complex shape compared to in-situ formed cloud at cirrus altitudes. I suggest to mention these in your text that your results are mostly representative for liquid origin clouds and may not be meaningful for in-situ formed clouds.

*Thank you very much for these comments. We have added text on different cirrus cloud types in section 2.3 as you suggest, and mentioned the significance of the role of liquid origin cirrus on cirrus microphysics, including the Luebke and Wernli citations here.*

• Page 8, lines 220-223: The influence of cloud origin and dynamics on crystal size is in agreement with other studies. Here, I would recommend to cite the paper by Luebke et al. 2016.

*We have been glad to incorporate the Luebke reference.*

• Page 12, lines 351-352: I cannot follow your argumentation that you found sub- limated ice crystals at the cloud top due to entrainment. Usually, the cloud top is dominated by nucleation of crystals and there you have the coldest tempera- tures and highest relative humidity wrt. ice (see e.g. Spichtinger and Gierens 2009). Thus, to find sublimated ice crystals at cloud top seems to unrealistic. At least this argument needs more explanation, citations etc and also discussion with point above.

*We have added this valuable reference. Our inference was based in part on flight videos, where we have often seen ice particles just at and even slightly above the cirrus cloud-top, the boundary of which is often not very sharply delineated in video. It certainly makes sense that the cloud top region overall has the highest supersaturation, though we presume that some particles right near the boundary are likely subject to influence from relatively dehydrated air above and also perhaps influenced by radiative heating. We also added a new reference (Wall et al. 2020 and a bit more explanation and qualification to this inference). We look forward to paying close attention to this question in future particle collections.*

• Figure 2: What is the large "rock" in the upper left part of the SEM picture. Maybe you can mentioned this also in the text because it is very conspicuous.

*This is a good point, we have added our understanding of this "rock" as a machining remnant in the text (line 318-320).*

**Technical comments/suggestions:**

- **Units in the manuscript**
  In ACP usually all values and their units are given as SI base unit. For example you use the kt for the wind speed which should be given in meters per sec- ond (m/s). This unit is also recommended by the World Meteorological Organi- zation for reporting wind speeds. I would recommend to go through the entire manuscript and change all non SI units like miles etc. to appropriate SI unit.

  *All non-SI units have been changed to standard SI units.*

- Page 2, line 43: I suggest to cite also Sourdeval et al. 2018 to have one repre- sentative paper of cirrus properties using lidar/radar technique.

  *We are glad to include the Sourdeval reference here.*

- Page 7, line 178: "Hitachi SU5000 is employs a Schottky ", the word "is" is to much.

  *This typo has been corrected.*

- 7, line 181: Minus is missing at the temperature value. Should be ~-160°C

  *This typo has been corrected.*

- Page 10, line 288: Capitalize the "c" --> panel C.

  *This has been corrected; all figure panel labels and references in the text have been checked and corrected for consistency.*

- Page 12, line 334-335: Please use another word than categories, because the reader expect than particles to be sorted in specific categories which is not the case here. It is more like a list of all the findings. You should use e.g. topics or findings.

  *This is a very good suggestion; we have replaced the wording with "four broad themes" (line 382).*

- Page 13, line 385: "finer than" instead of "finder that"

  *This typo has been corrected.*

- Page 14, line 392: I think you mean 500nm instead of 500 microns

  *This typo has been corrected.*

- Figure 4: a) No scale, please add a scale here. b)-left and c)-left Scale not readable. b)-right and c)-right Table not readable, please enlarge or skip. Peak classification in the diagramm not readable, please enlarge Peak labels.

  *We have added a scale bar for Figure 4 and enlarged the EDS spectral table text and peak labels.*

- Figure Supl. 1-F (a) and 1-G (a-f): Scale not readable. Can please add the same gray shadow behind the scale as you did in the other pictures.

  *Supplement figure scales have been edited for readability.*

Atmospheric Chemistry and Physics

[revised manuscript text omitted]

---

## Referee Report (RR1)

**Review of the Revision: Captured Cirrus Ice Particles in High Definition by Magee et al.**

**Overall impression and rating**

Thank you for answering all my questions and concerns. The revised manuscript improved. I really like the new Section 2.5. "ICE-Ball upgrades in progress" and also thanks for putting the CFD calculation into the supplementary material, which is sufficient.

I still recommend to increase the size of the scale in Figure 4b and 4c, which is very hard to read and unreadable in a printed version.

I recommend publication in ACP after considering my last technical point.

---

## Referee Report (RR2)

Overall recommendation

The authors made substantial revisions based on the reviewer's comments and I believe that the manuscript now has a better shape. The overall quality of this manuscript satisfies the standard of Atmospheric Chemistry and Physics and the methods that were used in this study are solid ("seeing is believing"). I recommend this manuscript will be published on Atmospheric Chemistry and Physics. My minor comments are listed below.

In reply to author's reply:

1. It is good to see that authors pointed this out in section 2 and conclusion sections.

2. I hope to see the improved version of ICE-Ball and use in aerosol science.

3. The authors defined several scales as nanoscale, 1-100 nm; mesoscale, 100 nm – 1 µm; and microscale, 1 µm – 500 µm. It is strange that defined mesoscale is smaller than microscale although "meso" means "middle or between". As the authors are aware that mesoscale is larger than microscale in meteorology.

12. A development of an automatic habit classification for the ICE-Ball data should be added on your to do list.

---

## Author Response (AR2)

February 5, 2021

Greetings,

Thank you very much to Dr. Krämer and the two reviewers of this manuscript, "Cirrus Crystals in High Definition", acp-2020-486.   We are very happy to receive word of the ACP publication acceptance.  All of your comments and suggestions have been very helpful.

With respect to several final technical comments and suggestions from the two reviewers, we appreciate both points and have gladly incorporated several small corrections to address these items.

Specifically:

1.  With respect to potential nomenclature confusion between "mesoscale meteorology" and the intermediate mesoscale size range between nanoscale and microscale, we have adjusted the language to "mesoscopic" instead of "mesoscale".  The "mesoscopic" term is a standard usage in condensed matter physics and should help avoid unintended association with mesoscale meteorology.
    (e.g. https://en.wikipedia.org/wiki/Mesoscopic_physics).

2.  We have increased the size of the scale bars on the panels Fig. 4b and 4c for improved readability.

The manuscript with the two changes above has been uploaded in .pdf and .docx format.  The 5 figure files have been saved in high-resolution, high quality .png format and uploaded in a single .zip archive.  The supplement files are unchanged.

Sincerely,

Nathan B. Magee, and co-authors